# First-in-human robotic supermicrosurgery using a dedicated microsurgical robot for treating breast cancer-related lymphedema: a randomized pilot trial

Tom J.M. van Mulken[1], Rutger M. Schols[1]*, Andrea M.J. Scharmga[1], Bjorn Winkens[2], Raimondo Cau[3], Ferry B.F. Schoenmakers[3], Shan S. Qiu[1], René R.W.J. van der Hulst[1] & MicroSurgical Robot Research Group

Advancements in reconstructive microsurgery have evolved into supermicrosurgery; connecting vessels with diameter between 0.3 and 0.8 mm for reconstruction of lymphatic flow and vascularized tissue transplantation. Supermicrosurgery is limited by the precision and dexterity of the surgeon's hands. Robot assistance can help overcome these human limitations, thereby enabling a breakthrough in supermicrosurgery. We report the first-in-human study of robot-assisted supermicrosurgery using a dedicated microsurgical robotic platform. A prospective randomized pilot study is conducted comparing robot-assisted and manual supermicrosurgical lymphatico-venous anastomosis (LVA) in treating breast cancer-related lymphedema. We evaluate patient outcome at 1 and 3 months post surgery, duration of the surgery, and quality of the anastomosis. At 3 months, patient outcome improves. Furthermore, a steep decline in duration of time required to complete the anastomosis is observed in the robot-assisted group (33–16 min). Here, we report the feasibility of robot-assisted supermicrosurgical anastomosis in LVA, indicating promising results for the future of reconstructive supermicrosurgery.

[1] Department of Plastic, Reconstructive and Hand Surgery, Maastricht University Medical Center, Maastricht, The Netherlands. [2] Department of Methodology and Statistics, Care and Public Health Research Institute (CAPHRI), Maastricht University, Maastricht, The Netherlands. [3] Department of Medical Robotic Technologies, Eindhoven University of Technology, Eindhoven, The Netherlands. A full list of consortium members appears at the end of the paper. *email: rutger.schols@mumc.nl

As a result of evolving technology in microscopes and instruments, microsurgeons are now able to perform supermicrosurgery by connecting vessels with a diameter between 0.3 and 0.8 mm for the reconstruction of lymphatic flow and vascularized tissue transplantation. Nonetheless, performance is limited by precision and dexterity of the human hands. Facilitating supermicrosurgical procedures by robot assistance could overcome these human limitations. A widely used robotic system for many different surgical disciplines is the Da Vinci system (Intuitive Surgical Inc.™, Sunnyvale, USA). This device was created to perform mainly endoscopic surgery and therefore showed limitations in microsurgical procedures[1–5].

A robotic system exclusively designed for reconstructive microsurgery has not been available up to now. Recently, microsurgeons of Maastricht University Medical Center (MUMC+, Maastricht, The Netherlands) in collaboration with technical engineers from Eindhoven University of Technology (TUe, Eindhoven, The Netherlands) have developed world's first dedicated robotic platform for (super)microsurgery, MicroSure's MUSA (MicroSure, Eindhoven, The Netherlands). The MUSA is designed to aid in stabilizing movements of the microsurgeon by filtering tremors and scaling down motions. The robot is easily maneuverable, equipped with arms holding genuine (super) microsurgical instruments that are easily placed into the holders, and are compatible with conventional surgical microscopes (see Fig. 1). Preclinical tests of the MUSA have confirmed the safety and feasibility of this robot in performing microsurgical anastomosis[6,7].

Owing to advances in early diagnosis and more effective treatments, breast cancer has become a chronic condition rather than a life-threatening illness. Therefore, special attention is paid for long-term sequelae of this condition, such as breast cancer-related lymphedema (BCRL), which affects 29.4% of breast cancer survivors within 2 years after surgery[8]. Treatment includes decongestive therapy consisting of the use of compressive garments and manual lymph drainage, and surgical interventions such as lymphatico-venous anastomosis (LVA). During the LVA procedure, where high precision and manual dexterity are essential for the outcome of the surgery[9], a supermicrosurgical anastomosis between lymphatic collecting vessels and the subcutaneous venous system is performed. It has been shown to be effective in reducing lymphedema severity in early-stage BCRL, thereby improving the quality of life (QoL) of the patient[10–13].

We report on the first-in-human use of the MUSA for robot-assisted LVA in patients suffering from BCRL. A randomized pilot study investigates patient outcome, duration of surgery, and quality of anastomosis in robot-assisted or manually treated female patients.

## Results

**Patient characteristics**. Twenty females were randomized to undergo robot-assisted or manual LVA. Two subjects from the robot-assisted group received manual LVA due to perioperative technical setup error of the MUSA. In one case, incidental damage of the microsurgical instruments occurred preoperatively in the sterilization process before the actual installation in the robot. The operating surgeon decided to proceed with the procedure, but perform the operation manually with an other microsurgical set that was not prepared for use in the robot. The other case consisted of a software initializing error of the robot, which was corrected and did not occur during the remainder of the study. In both cases the patients experienced neither harm nor inconvenience. Analyses were based on the final type of surgery conducted. Table 1 presents the baseline characteristics of patients receiving robot-assisted vs. manual LVA. Statistical

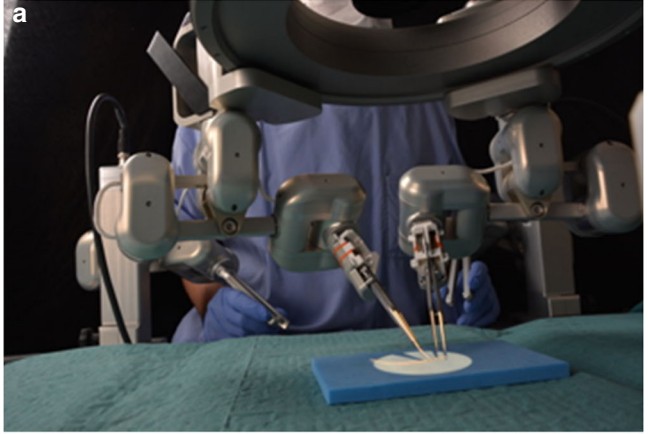

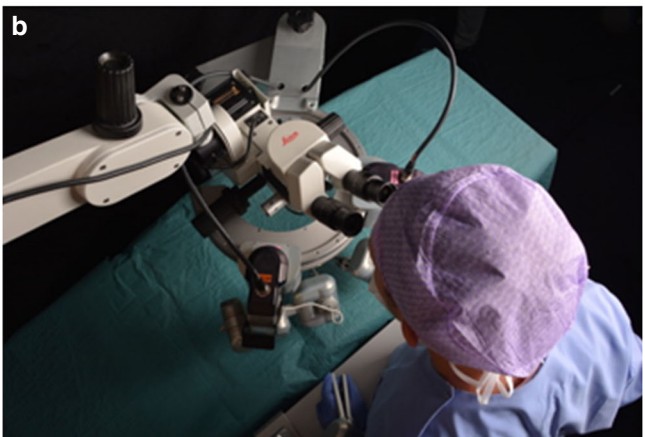

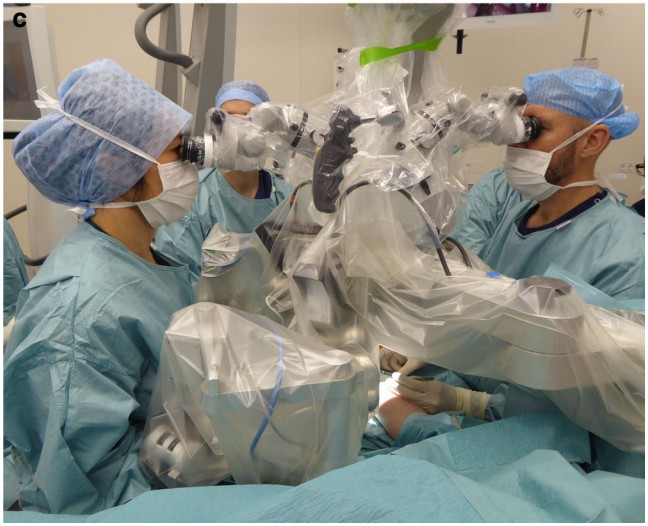

analyses showed no significant differences between the two groups.

**Robot-assisted vs. manual LVA**. Two microsurgeons, blinded for type of surgery, independently scored the quality of each anastomosis ($n = 40$), using the Structured Assessment of Microsurgery Skills (SAMS) and University of Western Ontario Microsurgical Skills Acquisition Instrument (UWOMSA) scoring methods[14,15]. As described in the Methods section, regarding both scoring systems, a higher score indicates better performance. Intraclass correlation coefficient (ICC) based on the mean SAMS score of all items per reader was fair, ICC 0.28, 95% confidence interval (CI) −0.02 to 0.53. ICC of the UWOMSA per domain

**Fig. 1 The microsurgical robot. a, b** Setup of the robot in a laboratory setting. In general the system is composed of the following: (1) master manipulators that are forceps-like joysticks, mounted to the operating table. These master manipulators are controlled by the operating surgeon. (2) A suspension ring that is attached to the operating table. The ring is placed between the operating field and the surgical microscope. (3) Slave manipulators that are robotic arms, which are attached to the suspension ring. The robotic arms can be equipped with genuine (super)microsurgical instruments. (4) Foot pedals that activate the system. A digital interface converts the movements of the master manipulators onto movements of the robotic arms. Motion scaling and tremor filtration can be adjusted by the software and controlled by foot pedals. **c** The MUSA in a clinical setting (the authors have preoperatively obtained patient's consent to publication of the image). The microsurgeon on the left controls the MUSA via two master manipulators, which are mounted to the operating table. Two slave manipulators, mounted to the suspension ring between the operating field and the surgical microscope, then mimic the surgeon's hand movement. In this case the microsurgeon on the right provides manual assistance during the procedure in an identical way as would be in conventional microsurgery cases with two surgeons.

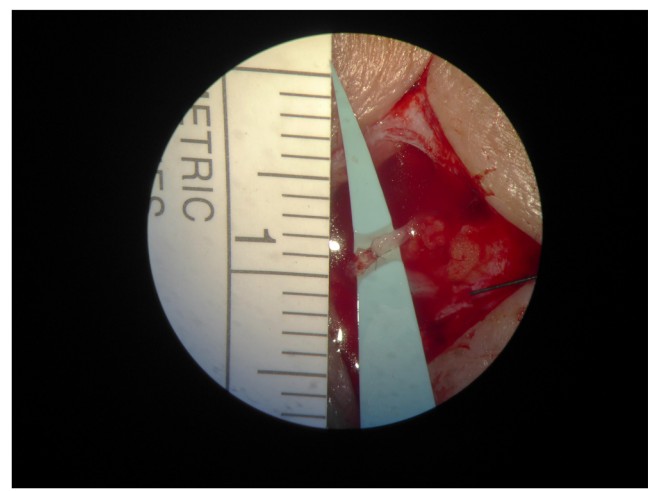

**Fig. 2 Microscopic view of a lymphatico-venous anastomosis.** View through the microscope on a completed LVA.

**Table 1 Patient characteristics.**

| | Robot-assisted LVA (n = 8) | Manual LVA (n = 12) | p-Value |
|---|---|---|---|
| Age | 60 ± 11 years | 60 ± 7 years | p = 0.85 |
| BMI | 27 ± 4 kg/m² | 25 ± 5 kg/m² | p = 0.31 |
| Current smoking | 0 | 0 | |
| Mean years of lymphedema | 4.75 ± 3.49 years | 9.25 ± 9.14 | p = 0.20 |
| ISL classification | | | |
| Stage 1 | 1 | NA | |
| Stage 2a | 7 | 11 | |
| Stage 2b | NA | 1 | |

Data are shown as mean ± SD or absolute number.
*Lymph-ICF* Lymphedema Functioning, Disability, and Health questionnaire, *ISL* International Society of Lymphology, *NA* not applicable, *UEL* upper extremity lymphedema index.
Reported *p*-values were obtained from independent samples' *t*-test, where Mann–Whitney *U*-tests showed similar results.
Source data are provided as a Source Data file.
Patient characteristics of patients receiving robot-assisted vs. manual LVA.

was also fair; preparation ICC 0.21, 95% CI −0.07 to 0.48, suturing ICC 0.37, 95% CI 0.01 to 0.64 and final product ICC 0.35, 95% CI −0.10 to 0.68. The mean SAMS score was significantly higher in the manual LVA group compared with the robot-assisted group: 4.0 ± 0.3 and 3.2 ± 0.4, respectively ($p < 0.001$). The mean UWOMSA score was also significantly higher in the manual LVA group compared with the robot-assisted group: 4.0 ± 0.5 and 3.4 ± 0.3, respectively ($p < 0.001$). Supplementary Table 1 shows the mean SAMS and UWOMSA scores per item (SAMS) and domain (UWOMSA).

In total, 14 anastomoses were completed using robot assistance ($n = 8$ patients, mean 1.75 ± 0.5, range 1–2) and 26 anastomoses were performed manually ($n = 12$ patients, mean 2.1 ± 0.7, range 1–4). All completed anastomoses were patent. See Fig. 2 for an example of a completed LVA. Mean (±SD) duration and range of anastomosis in the robot-assisted group compared with the manual group was significantly different: 25 ± 6 min, range 16–33 min and 9 ± 6 min, range 4–36 min, respectively ($p < 0.001$). However, a steep decrease in duration of time required to complete the anastomosis was observed in the robot-assisted group (see Supplementary Fig. 1). Mean total time of the full surgical procedure accounted 81 min for the manual group and 115 min for the robot-assisted group.

Mean patients' convenience during the procedure in the robot-assisted group was 8.0 (±2.0, range 4–10) and 8.6 (±1.1, range 7–10) in the manual group. Surgeons' satisfaction of the procedure was 3.1 (±0.6, range 2–4) in the robot-assisted group and 3.8 (±0,8, range 2–5) in the manual group. See Supplementary Tables 2, 3, and 4. Source data are provided as Source Data File.

**Follow-up data.** Table 2 shows baseline, one month and three months data of daily use of compressive garment, manual lymph drainage, mean Lymphedema Functioning, Disability, and Health questionnaire (Lymph-ICF) total score and mean Upper Extremity Lymphedema (UEL) index of the affected arm. At 1 month and 3 months post surgery, data of some measurements were missing due to either no show at the outpatient clinic ($n = 1$ at 1 month, $n = 1$ at 3 months) or wearing compressive garment during visit resulting in an incorrect arm measurement ($n = 1$ at one month, $n = 3$ at three months). In case of no show, the Lymph-ICF questionnaire was sent by mail.

At 3 months follow-up, one patient in the robot-assisted group reported daily use of compressive garment (12.5%), whereas in the manual group 16.7% ($n = 2$) resumed daily use of compressive garment.

No significant differences in mean Lymph-ICF total score between the groups at 1 month and 3 months were observed. Supplementary Fig. 2 shows the mean scores per domain and mean total score of the Lymph-ICF questionnaire for the robot-assisted and manual LVA group. Between baseline and three months, mean percentage difference in Lymph-ICF total score decreased in both groups; −41.17 in the robot-assisted group vs. −41.57 in the manual LVA group ($p = 0.98$). A linear mixed model analysis showcased that for Lymph-ICF score no significant difference was found according to the intervention (manual vs. robot-assisted LVA) at 1 month (8.31, 95% CI −6.75 to 23.37, $p = 0.26$) and three months (0.69, 95% CI −13.41 to 14.51, $p = 0.92$), respectively. No significant differences in mean UEL index of the affected arm between the groups at 1 month and 3 months were observed. Furthermore, between baseline and 3 months, mean percentage difference in UEL of the affected arm was slightly decreased in the robot-assisted group and increased in the manual LVA group (−0.93 and 0.36, respectively, $p = 0.66$).

Linear mixed model analysis showed that for UEL index, no significant intervention effect (manual vs. robot-assisted LVA), at

**Table 2 Baseline, one 1, and 3 months follow-up.**

| | Baseline | | | One month | | | | | Three months | | | | |
|---|---|---|---|---|---|---|---|---|---|---|---|---|---|
| | Robot LVA (n = 8) | Manual LVA (n = 12) | p-Value | Robot LVA (n = 8) | Manual LVA (n = 12) | Estimate | 95% CI | p-Value | Robot LVA (n = 8) | Manual LVA (n = 12) | Estimate | 95% CI | p-Value |
| Daily use of compressive garment | | | | | | | | | | | | | |
| Yes | 7 | 11 | | 1 | 1 | | | | 1 | 2 | | | |
| No | 1 | 1 | | 6 | 10 | | | | 7 | 9 | | | |
| Missing | 0 | 0 | | 1 | 1 | | | | 0 | 1 | | | |
| Manual lymph drainage | | | | | | | | | | | | | |
| Yes | 7 | 7 | | 2 | 2 | | | | 7 | 9 | | | |
| No | 1 | 5 | | 5 | 9 | | | | 1 | 2 | | | |
| Missing | 0 | 0 | | 1 | 1 | | | | 0 | 1 | | | |
| Mean lymph-ICF total score | 38 ± 16 | 49 ± 16 | 0.17 | 28 ± 17 | 29 ± 21 | 8.31 | −6.75 to 23.37 | 0.26 | 22 ± 16 | 29 ± 19 | 0.69 | −13.14 to 14.51 | 0.92 |
| Missing | | | | 0 | 1 | | | | 1 | 0 | | | |
| Mean UEL index affected arm | 116 ± 24 | 122 ± 20 | 0.57 | 114 ± 24 | 125 ± 20 | −3.95 | −10.62 to 2.75 | 0.23 | 113.01 ± 21 | 125 ± 19 | −0.33 | −6.69 to 6.03 | 0.91 |
| Missing | | | | 2 | 1 | | | | 2 | 1 | | | |

Data are shown as absolute number or mean ± SD.
Baseline, 1 month, and 3 months follow-up of patients receiving robot-assisted vs. manual LVA. Difference of Lymph-ICF and UEL index between robot-assisted and manual LVA at baseline were analyzed using independent samples t-test. Lymph-ICF and UEL index at 1 and 3 months were analyzed using linear mixed models, corrected for baseline. Source data are provided as a Source Data file.

1 month (−3.95, 95% CI −10.62 to 2.75, p = 0.230) and 3 months (−0.33, 95% CI −6.69 to 6.03, p = 0.914) was found.

**Adverse events**. In the robot-assisted group, two patients reported a suspected erysipelas infection for which oral antibiotics had been administered; one of these patient had already suffered from recurrent erysipelas infections in the past. No serious adverse events were reported.

## Discussion

Robot assistance may potentially overcome the human limitations in challenging supermicrosurgical interventions such as the LVA procedure in BCRL. Recently, a dedicated microsurgical robot for (super)microsurgery—the MUSA—was developed and showed safety and feasibility in a preclinical study. We currently investigated and proved the feasibility of performing robot-assisted LVA using the MUSA in a randomized in-human pilot study.

The Da Vinci system (Intuitive Surgical, Inc.™, Sunnyvale, US) was the first robotic device with Food and Drug Administration approval and is particularly designed for minimally invasive surgery. The system offers a three-dimensional stereoscopic vision, instruments with six degrees of motion freedom, scalable movements, and elimination of tremor. Previous studies have reported the feasibility of using robot assistance for various microsurgical procedures in surgical specialties[1,4,5,16–19]. Despite the aforementioned robotic advances, experimental studies observed drawbacks when using the Da Vinci system in microsurgical procedures[1–5]. In reconstructive microsurgery, refined instruments of small size for subtle handling are mandatory. The surgical instruments of the Da Vinci are large and powerful for microsurgery. In addition, visual magnification is limited and resolution is poor at high levels of magnification, making (super) microsurgical procedures difficult.

The MUSA is a lightweight, small-sized system, which can be mounted to the surgical table and microscope, providing easy integration in the operating theatre with minimal adaptations of the theatre layout and organizational planning. Key features of the MUSA are application of motion scaling and tremor filtering to enhance microsurgical performance. Moreover, the robot is compatible with standard surgical microscopes and microsurgical instruments.

In the current pilot study it was shown that it was feasible to complete supermicrosurgical anastomoses in patients with BCRL using the MUSA. Total time of the surgical procedure for both manual and robot-assisted LVA remained within 115 min, which is fairly reasonable for a procedure performed under local anesthesia. Despite a longer time to perform the anastomosis in the robot-assisted group, a steep decline in duration of anastomosis performed by one single microsurgeon, was already observed during the trial period. In light of this, literature showed that using the Da Vinci robotic system for microvascular anastomosis, the learning curve accounted 22 trials with an average starting anastomosis time of 101 ± 30 min and an estimated mean learning plateau of 33 ± 15 min[20]. With an average anastomosis time of 25 ± 6 min after eight robot-assisted LVA procedures, the MUSA shows already good performance.

In previous studies by Cornelissen et al.[21] and Winters et al.[22], it was shown that QoL improved after manual LVA in females with BCRL 1 year after surgery. The current study confirms this improvement of QoL (as assessed by Lymph-ICF score) in both groups (robot-assisted and manual) at three months follow-up. UEL index improved slightly in the robot-assisted group.

The aforementioned study by Cornelissen et al.[21] also showed that 85% of the study population discontinued wearing compressive garments after 1 year. In the current study 87.5% of the

robot-assisted group and 83.3% of the manual group discontinued wearing their sleeve daily at three months. Longer follow-up is needed to compare 1-year outcome.

Mean SAMS and UWOMSA scores were significantly higher in the manual LVA group compared with the robot-assisted LVA group. This also accounts for the individual scores of manual versus robotic as outlined in Supplementary Table 1. In addition, both surgeons' satisfaction and patients' convenience during the procedure were slightly higher in the manual group compared with the robot-assisted group. However, the current report of the feasibility of robotic supermicrosurgery consists of small groups. Based on these small groups of robotic vs. manual, we cannot yet make solid conclusions relating the observed differences between the two surgical approaches. This matter will certainly be discussed in future larger studies. The aforementioned finding regarding SAMS scores is in line with previous preclinical experiments[6]. It might be suggested that the use of SAMS is less appropriate for assessing robot-assisted microsurgical procedures. Selber et al. developed a modified SAMS, the Structured Assessment of Robotic Microsurgical Skills (SARMS), including six additional items such as camera movement, depth perception, wrist articulation, atraumatic tissue handling and atraumatic needle handling[23] developed a modified SAMS, the Structured Assessment of Robotic Microsurgical Skills (SARMS), including six additional items such as camera movement, depth perception, wrist articulation, atraumatic tissue handling and atraumatic needle handling. This SARMS score cannot be used in supermicrosurgery as the currently available cameras in any robotic system lack the high magnification and modern operation microscopes are used. Therefore, in the design of the current study with the MUSA and manual supermicrosurgical LVA, both using the same operation microscope for high magnification, the SAMS was selected. In addition, the UWOMSA was used.

The use of the MUSA made double blinding not possible, neither for the surgeon, nor for the patient as the procedures were wide-awake under local anesthesia. We believe the effect of (performance) bias was still minimal, as the results showed that the subjectively filled-out Lymph-ICF questionnaire scores improved in both groups and that objectively measured UEL index improved in the robot-assisted group.

In summary, our data provide the first-in-human proof that robot-assisted supermicrosurgical LVA in patients suffering from BCRL is in fact feasible and safe using the MUSA. Improvement of QoL and a decrease in arm volume was detected at three months follow-up. Due to the limited series of patients no significant intervention effect could yet be detected. Moreover, despite a longer time to perform the anastomosis in the robot-assisted group, a steep decline in time required to create an anastomosis was already observed in the robot-assisted group indicating promising results for the future. Assessment of reproducibility amongst other surgeons is necessary since the anastomosis in our study were only performed by one experienced microsurgeon. Larger multicenter trials, with involvement of multiple trained microsurgeons, are needed to confirm our data and further elucidate the therapeutic value of using the MUSA in (super)microsurgery.

## Methods

**Patients**. Ethical approval was obtained from the Institutional Review Board of the academic hospital Maastricht/Maastricht University (Maastricht, the Netherlands). The study was registered in the Netherlands Trial Register (NTR6291: https://www.trialregister.nl/trial/6291). Procedures followed were in accordance with the Declaration of Helsinki.

Between June 2017 and December 2017 females suffering from unilateral BCRL referred to the lymphedema outpatient clinic at MUMC+ were invited to participate in this randomized pilot study. After this date recruitment of the pilot trial was closed. Eligible patients were female adults ≥18 years old, who suffered from unilateral early-stage lymphedema of the arm (Stage 1 and 2 of the International Society of Lymphology (ISL) classification, mild, persistent, or fibrotic lymphedema[24]) after breast cancer treatment with axillary lymph node surgery and/or radiotherapy, and at least underwent three months of complex decongestive therapy without symptoms alleviation. Patients were excluded from this study if they presented distant breast cancer metastases, current substance abuse, known indocyanine green (ICG) allergy, a previous LVA in the affected arm or a non-viable lymphatic system as determined by near-infrared fluorescence (NIRF) imaging using ICG[25,26]. Written informed consent was obtained from all eligible subjects and block randomization (block size 4) by a computer-generated list was used to allocate the subjects[27]. The authors have preoperatively obtained patient´s consent for publication of identifiable images.

Following standard preoperative screening for LVA, NIRF lymphography using ICG to map the superficial lymphatic system was conducted. ICG dye (0.03–0.05 ml of 5 mg/ml concentration ICG, VERDYE 25 mg, Diagnostic Green, Germany) was injected intradermally in the second and fourth web spaces in the hand. Using NIRF lymphography, near-infrared light emitted by the dye was detected, thereby visualizing lymphatic collecting vessels, their functionality, and level of occlusion (see Fig. 3a). Preoperative markings indicating the site for incision were photographed along with a measuring tape (Fig. 3b). Furthermore, study subjects completed the Lymph-ICF, a questionnaire consisting of 29 questions based on function, activity limitations and participation restrictions divided into five domains (physical function, mental function, household activities, mobility, and social life aspects)[28]. Items are scored on a Visual Analog Scale (VAS) from 0 to 100 mm. Higher scores indicate worsening of the QoL. Last, UEL index, circumference measurement of five points on the affected arm corrected for body mass index (BMI), was calculated[29]. A decrease in arm circumference results in a lower UEL index. The same individual took these measurements preoperatively, 1 month and 3 months after surgery.

**Microsurgical platform**. The MUSA has been evaluated in preclinical tests of the system[6,7]. In general, the system is composed of (see Fig. 1):

1. Master manipulators manipulated by the surgeon using forceps-like joysticks, mounted to the operating table.
2. A suspension ring that is placed between the operating field and the surgical microscope. The suspension ring is attached to the operating table and can carry multiple robotic slave arms.
3. Slave manipulators, which are robotic arms that can be equipped with genuine microsurgical instruments.
4. Foot pedals that can activate the system and control motion scaling.

Using the master manipulators, one or two operating surgeons can control the activated robotic slave arms. Through the digital interface, tremor filtration and motion scaling provides enhanced precision of the surgeons' motion. Altogether, with this setup, the surgeons can remain seated close to the patient and have a direct view of the patient and the operation site. This enables a quick switch between a conventional manual approach and robot assistance in microsurgery cases. This possibility of hybrid operations allows robot assistance only to be used in phases of the procedure where high precision is needed.

The system can be combined with genuine microsurgical instruments and conventional surgical microscopes. We use sterile adapters, which can easily be coupled to the robot and can be loaded with genuine microsurgical instruments. See Fig. 1C. Normal surgical workflow is not disrupted by installation and sterile draping of the MUSA, which can be performed prior to the patient coming into the operating theatre.

**Surgical procedure and data recording**. One experienced microsurgeon (SQ; with 11 years of experience in plastic surgery, 550 free flaps, and 250 (super) microsurgical LVA procedures) performed all LVA procedures of the enrolled participants. Before initiation of the clinical study, the microsurgeon had 20 h of training on the MUSA; she completed at least three anastomoses with a SAMS score of at least 3.

Photographs of the preoperatively performed NIRF lymphography at the outpatient clinic indicated the incision sites. Local anesthesia was used for all the procedures. A mix of marcaine and epinephrine (2.5 mg/mL, 5 mcgr/mL) was preferred. Incisions of 1.5–2.0 cm were made at the predetermined sites and dissection of the subdermal region took place to identify viable lymphatic vessels and venules. This first part was performed manually. One or more anastomoses were performed, either robot-assisted or manually depending on the randomization. After transection, end-to-end anastomoses were completed between the lymphatic vessel and recipient venule with an 11-0 ethilon suture (Ethicon, Johnson & Johnson, USA). The patency of the anastomosis was checked by observing the blood pass from the venule through the anastomosis in the lymphatic collecting vessels or lymphatic flow through the anastomosis. The wound was closed using interrupted transcutaneous sutures with 4-0 ethilon and light compressive dressings were placed over the wound. These procedures were recorded on digital video through the microscope (ZEISS OPMI PENTERO 900, Carl Zeiss Meditec AG, Germany). Duration of anastomosis was recorded. Patients were allowed to perform their daily activities directly after the surgery, but were not allowed to wear compressive garments or receive manual lymph drainage 4 weeks postoperatively. This is the standard care for patients undergoing LVA procedures in our center.

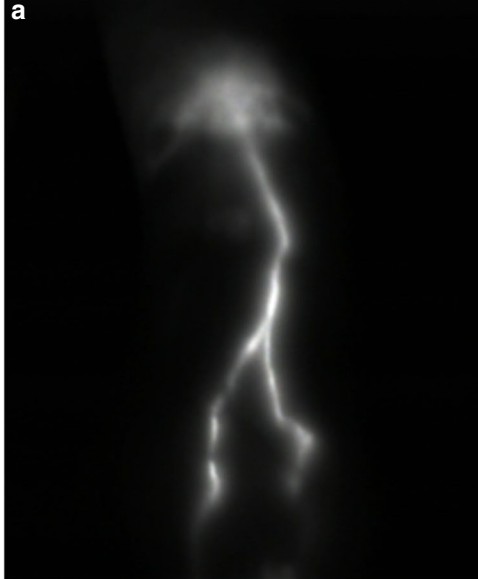

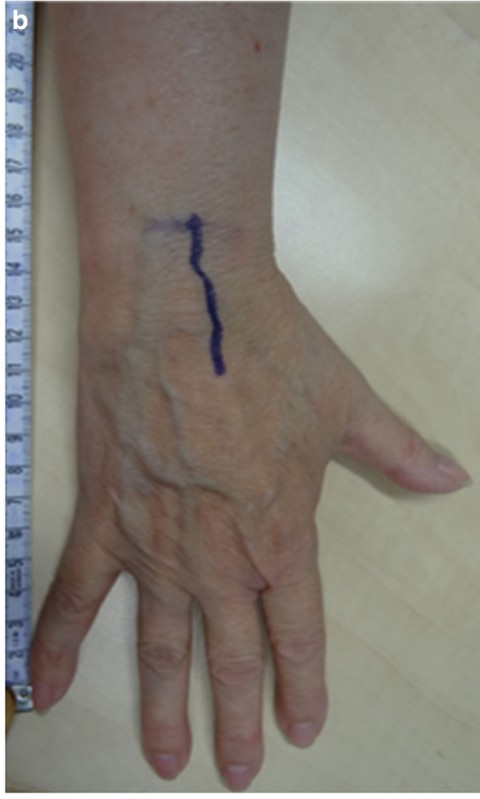

**Fig. 3 Preoperative NIRF lymphography and corresponding markings. a**
An example of preoperative NIRF lymphography after intradermal ICG
administration in the second and fourth finger web spaces of the right hand
of a study subject, as performed in the lymphedema outpatient clinic. **b**
Corresponding preoperative markings based on findings of NIRF
lymphography in the same patient. Measuring tape is used to indicate the
site for incision during the actual LVA procedure.

**Study parameters**. The following patient characteristics were obtained: age, BMI,
ISL stage, smoking status at baseline, daily use of compression garment, and
manual lymph drainage at baseline, one month and three months post surgery.
Primary outcome regarding the surgical procedure was quality of the anastomosis.
Secondary outcomes included duration of the surgery, problems or technical errors
during the procedure, postoperative complications and adverse events, Lymph-ICF
total score (i.e., patient's QoL), and UEL index of the affected arm (i.e., arm
volume), surgeons' satisfaction and patient's satisfaction with respect to the

procedure were recorded. Total time of the surgical procedure was defined as start
of incision until end of the last skin suture. Total time of anastomosis was defined
as start of the first microsurgical suture until completion of the last suture of the
anastomosis. Quality of each anastomosis was scored independently and blinded
for type of surgery by two experienced microsurgeons using the validated SAMS
method[14] and UWOMSA score[15]. The SAMS score contains twelve separate items,
scored from 1 (bad) to 5 (excellent), grouped into four areas (dexterity, visuo-
spatial ability, operative flow and judgement), each subdivided into three technical
components[14]. Also, overall performance (1 bad–5 excellent) and indicative skill
(1 novice–5 expert) were rated[14]. The UWOMSA evaluates microsurgical com-
petency using three items; preparation, suturing and final product, rated on a
five-point Likert scale, highest score 5[15]. At the end of each operation the patients'
overall convenience was assessed using a ten point VAS score. The higher the score,
the more convenient the patient had experienced the surgery. In addition, the
surgeons' satisfaction with the procedure was assessed using a five-point score; the
higher the score, the higher the satisfaction of the surgeon.

Serious adverse events resulting in death, a life-threatening condition,
hospitalization or a persistent disability, were to be recorded in the case report
form. Adverse events defined as any undesirable experience occurring to a subject
during the study, e.g., infection with consequently antibiotics, were also reported.

**Statistical analysis**. Baseline characteristics were analysed (age, BMI, smoking,
daily use of compressive garments, manual lymph drainage, ISL stage, lymph-ICF
total score, and UEL index of the affected arm) for both the robot-assisted and
manual LVA group. Numerical variables were reported as mean and SD, and
categorical variables as percentages or absolute numbers. Because of the small
sample size in both groups, the independent Student's $t$-test and the
Mann–Whitney $U$-test were both evaluated for differences between groups for the
variables age, BMI, UEL index of the affected arm and lymph-ICF total score.
Differences per patient (percentage) were calculated for UEL index of the affected
arm and lymph-ICF total score between baseline and three months post surgery.

In addition, linear mixed model analyses with an unstructured covariance
structure for repeated measures were used to evaluate whether intervention (robot-
assisted or manual LVA) influenced lymph-ICF total score or UEL index of the
affected arm over time (one month or three months post surgery, after correction
for baseline differences).

For the quality of the anastomosis assessed using SAMS and UWOMSA,
interreader reliability of two readers was analyzed for the mean SAMS score per
reader for all items and UWOMSA score per domain using ICC with a two-way
random effect model with absolute agreement, single measures. Reliability was
rated according to Landis et al.; <0.00 poor, 0.00–0.20 slight, 0.21–0.40 fair,
0.41–0.60 moderate, 0.61–0.80 substantial, 0.81–1.00 almost perfect[30]. SAMS and
UWOMSA were further analyzed based on the average scores of both readers.

Differences in duration of anastomosis, and SAMS and UWOMSA score of the
anastomoses between robot-assisted or manual LVA were analyzed using an
independent samples' $t$-test. As sensitivity analyses, these differences were also
analyzed using linear mixed models with surgery type as fixed factor and a random
intercept on patient level to account for the clustering of anastomoses within patients.

Differences in patient comfort and surgeon performance between robot-assisted
or manual LVA were analyzed using an independent samples' $t$-test.

Statistical analyses were performed using SPSS Statistics for Windows version
23.0 (IBM Corp., Armonk, NY, USA). A $p$-value of <0.05 was considered
statistically significant.

**Reporting summary**. Further information on research design is available in
the Nature Research Reporting Summary linked to this article.

## Data availability
The authors declare that all data supporting the findings of this study are available within
the paper and its supplementary information. Source data are provided as a Source Data file.

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

## Acknowledgements

We acknowledge the OR-staff and the medical technology department of Maastricht University Medical Center for their support. The robotic equipment for this pilot trial was provided by MicroSure.

## Author contributions

T.J.M.v.M.: plastic surgeon, study design, clinical translation of the MUSA, manuscript writing, and review. R.M.S.: plastic surgeon in training, principal investigator, study design, participant recruitment, manuscript writing, and review. A.M.J.S.: post-doc researcher robotic microsurgery, study design, participant recruitment, data recording, manuscript writing, and review. B.W.: statistician, statistical assistance, and manuscript review. R.C. and F.S.: robotics/software engineers, design and constructions of the MUSA, and manuscript review. S.S.Q.: microsurgeon, study design, performed all L.V.A. procedures, manuscript writing, and review. R.R.W.J.v.d.H.: plastic surgeon, study design, clinical translation of the MUSA, and manuscript review. MicroSurgical Robot research group: data recording and manuscript review.

## Competing interests

The authors declare the following competing interests: R.M.S., A.M.J.S., B.W., F.S., S.S.Q. MicroSurgical Robot research group—no relevant conflict of interest. T.J.M.v.M., chief medical officer at MicroSure (shareholder). R.R.W.J.v.d.H., shareholder in MicroSure. R.C., chief technical officer at MicroSure (shareholder).

## Additional information

## MicroSurgical Robot Research Group

Xavier H.A. Keuter[1], Thomas M.A.S. Lauwers[1], Andrzej A. Piatkowski[1], Juliette E. Hommes[1], Dionne S. Deibel[1], Jessie E.M. Budo[1], Jai Scheerhoorn[1] & Maud E.P. Rijkx[1]

