## [Peer Review File · Nature Communications]

Reviewers' comments:

Reviewer #1 (Remarks to the Author):

This is important new knowledge that will raise awareness of the feasibility of treating lymphedema and that supermicrosurgery is a clinical advancement that has progressed from the laboratory to the operators.

Some suggestion for phrasing: "decline in duration of anastomosis was observed" can be reworded as "duration of time required to complete the anastomosis " - otherwise it may be mistaken for duration till failure of anastomoses. But this is a linguistic nuance.

Figure S1 could benefit from a mean bar for the robot and manual group
The authors should explain SAMS score and UWOMSA scores earlier in the text- is higher score better or worse should be elucidated before the tables and figures are presented with the scores.

Table 2 should include asterisks for any columns that show statistical difference at the $p=0.05$ value (they have included 9% CI which are defacto the same - but it will be easier for the reader)

Reviewer #2 (Remarks to the Author):

I would like to commend the authors for their novel method of performing lymphatico-venous anastomoses in a cohort of 20 patients with ISL stages I and II BCRL using a novel robotic platform. In their manuscript, the authors detail their initial experiences using the MicroSure Robot and report on variables including baseline patient characteristics, number of anastomoses performed, objective evaluation of anastomoses, operative time, anastomotic time, and 1 and 3-month patient outcomes.

There are a few areas where I believe this work can be strengthened.

The authors detail that the aim of their study is to "prove the feasibility of robot-assisted super microsurgical anastomosis in LVA...future of reconstructive microsurgery." Two patients were converted from the robotic cohort to the manual cohort - as the emphasis is surgical feasibility, the reader would be very interested in learning more about the issues that developed in these two cases (which represents 20% of their study group) and how the issues were or were not addressed.

The authors state that a single senior surgeon performed all LVA's (both robotic and manual) in the online methods section. If this is the case, the provided generalizations about the learning curve for this robotic platform may be rewritten to reflect that they are specific to one single surgeon with (unknown/not provided) experience performing LVA's. This would further allow for future assessment of reproducibility amongst other surgeons of similar experience and seniority.

An important point of clarification: Does the time needed to perform LVA using the robotic platform include time for robot setup, placement, and calibration? If not, this is an important point worthy of inclusion/discussion.

The authors employ various tools for assessment which is commendable. The authors describe a high percentage of patients who discontinued their compression sleeve at three months. However, the authors also state that patients were "not allowed to wear compressive garments or receive manual lymphatic drainage four weeks post-operatively." What is the standard of care for patients following this 4-week time point regarding compression therapy? Can the authors provide more

information regarding their rationale for cessation of compression and detail their standard post-operative compression protocol for LVA? Without this information, statistics reported regarding discontinuation of garment use in both groups can be misleading. Moreover, were surveillance measurements taken by the same individuals at each time point?

It would be helpful to have more information regarding the overall standard protocol for patient surveillance at the institution for patients undergoing LVA.

The inclusion of two tools to validate anastomoses is commendable. As a feasibility study, information regarding the time needed for raters to complete their assessments would be informative for other institutions interested in this robotic platform and evaluative techniques used (including barriers faced).

The authors are to be commended on the first-in-human study of robotic-assisted super microsurgery. This manuscript details a successful attempt by a single surgeon to use a novel platform in performing LVA's with generally similar outcome measures between groups. Further the authors include the multiple outcome metrics used to assess: anastomoses number and quality, operative time, and patient outcome measures used. Additional information is needed for the reader to assess whether or not this novel platform can be implemented on a broader scale with similar success.

Reviewer #3 (Remarks to the Author):

I would like to start by congratulating the authors on their paper. It is well written, addresses an important need, and as with all new technologies, authors can be excused for being overly enthusiastic about their new technology.

I will start with more general comments and end with some specifics.

Given that this is a first study in human, and it is a comparison between manual versus robotic surgery, the difference between the two approaches needs a more elaborate discussion. The differences are not just limited to surgical time, or overall scores. In supplementary table 1 (which should be part of the main article), we note that the patency of robotic anastomosis is less than that of manual surgery, that bleeding control is less and of equal importance, the number of "achieved" anastomosis per patient was higher for manual rather than robotic procedures (robotic 1.75 vs manual 2.2). The importance of these differences between the two surgical approach should be discussed. Overall surgical time can as much be surgeon related (tiredness) or available OR time for the procedure.

I would also appreciate a more detailed report, even if subjective, on surgical situations where the robot is superior to manual surgery - with smaller, more delicate vessels? difficult angles of approach?

Patients had longer duration of edema and somewhat more advanced stages in the manual group, though these differences (small numbers) did not reach statistical significance. Nonetheless, an analysis that would have only considered patients with a stage 2a might provide a more homogeneous basis for a comparison of the two approaches. Does this amplify certain of the differences observed or the outcomes.

Also it is not clear how much training the surgeon had on the system prior to performing his first human surgery. I think it is important to state this explicitly in terms of hours, number of simulated cases, or sequences of an established training module, not just indicate that an

experienced surgeon performed the surgery.

Specific items:

Abstract page 3 line 60: a decline in the time needed to complete an anastomosis was observed... Also provide number indicating the difference between robotic and manual surgery

P.8: be more explicit about the technical errors that precluded robotic surgery in two patients.

P.9 line 183 - you state the number of anastomosis completed, were all patent or equally patent at the end of surgery? If not indicate the number of patent anastomosis or indicate total number of attempts.

Line 190 - what corresponds to "total time". Does it include installation and draping or only the surgical procedure?

P.10 Figure 2 does not add additional information, unless the image provides a representative image of a manual and a robotically completed anastomosis.

P.16 figure S2, please add error bars to your graph.

P.18 324 please discuss in more detail other parameters such as patency, number of anastomosis. Time constraints on the use of the operating room seem a more important factor than the time for achieving an anastomosis, given the difference in the number of achieved anastomosis between the two groups.

P.19 347 yes, but we are not interested in assessing the performance of the robot or optimizing its performance but in comparing outcome between robot and manual surgery, though a scale that assesses robotic performance will in the future be needed.

P.25 Given the journal and the novelty of your robot, I would give a more elaborate description, as well as an adequate description of the forceps on the robotic system. A clearer outline of whether this is a tele-operated robot, a co-manipulator, etc is needed. Also is there a remote center of motion? This can be provided as a supplement, at least the figures.

487; when you mention combined with genuine microsurgical instruments, you mean these can be fitted onto the robot? How much time does the installation and draping take. Is there no restriction to the surgical field or collaboration between assistant and head surgeons? Was this assessed during the trial?

491: as previously stated, how much training on the robot did the surgeon have?

Marc D. de Smet, MDCM, PhD, FRCSC, FMH

Response to the reviewers

Reviewer 1

Original text:

Reviewer #1 (Remarks to the Author):

This is important new knowledge that will raise awareness of the feasibility of treating lymphedema and that supermicrosurgery is a clinical advancement that has progressed from the laboratory to the operators.

Some suggestion for phrasing: "decline in duration of anastomosis was observed" can be reworded as "duration of time required to complete the anastomosis " - otherwise it may be mistaken for duration till failure of anastomoses. But this is a linguistic nuance.

Reply: We have adjusted the text accordingly on page 3 line 60, page 10, lines 221 and page 20 lines 43.

Figure S1 could benefit from a mean bar for the robot and manual group

The authors should explain SAMS score and UWOMSA scores earlier in the text- is higher score better or worse should be elucidated before the tables and figures are presented with the scores.

Reply: We have included a mean bar for both robot and manual groups. The current revised graph with mean time is included in the manuscript on page 13.

The authors should explain SAMS score and UWOMSA scores earlier in the text- is higher score better or worse should be elucidated before the tables and figures are presented with the scores.

Reply: Thank you for the suggestion. We have added the following sentence on page 11 lines 228-229: 'As described in the Methods section, regarding both scoring systems, a higher score indicates better performance.'

Table 2 should include asterisks for any columns that show statistical difference at the $p=0.05$ value (they have included 9% CI which are defacto the same - but it will be easier for the reader)

Reply: Unfortunately none of the p-values are statistically significant. Therefore, we cannot place any asterisks.

Reviewer 2

Original text:

I would like to commend the authors for their novel method of performing lymphatico-venous anastomoses in a cohort of 20 patients with ISL stages I and II BCRL using a novel robotic platform. In their manuscript, the authors detail their initial experiences using the MicroSure Robot and report on variables including baseline patient characteristics, number of anastomoses performed, objective evaluation of anastomoses, operative time, anastomotic time, and 1 and 3-month patient outcomes.

There are a few areas where I believe this work can be strengthened.

The authors detail that the aim of their study is to “prove the feasibility of robot-assisted super microsurgical anastomosis in LVA...future of reconstructive microsurgery.” Two patients were converted from the robotic cohort to the manual cohort - as the emphasis is surgical feasibility, the reader would be very interested in learning more about the issues that developed in these two cases (which represents 20% of their study group) and how the issues were or were not addressed.

Reply: Thank you for your comment. We have incorporated an explanation of the aforementioned two technical failures in the text (page 9 line 195-201).

In one case, incidental damage of the microsurgical instruments occurred preoperatively in the sterilization process before actual installation in the robot. The operating surgeon decided to perform the operation manually with another microsurgical set that was not prepared for use in the robot. The other case consisted of a software initializing error of the robot, which was corrected and did not occur during the remainder of the study. In both cases the patients experienced no harm or inconvenience.

The authors state that a single senior surgeon performed all LVA's (both robotic and manual) in the online methods section. If this is the case, the provided generalizations about the learning curve for this robotic platform may be rewritten to reflect that they are specific to one single surgeon with (unknown/not provided) experience performing LVA's. This would further allow for future assessment of reproducibility amongst other surgeons of similar experience and seniority.

Reply: We have added that the microsurgeon has 11 years of experience in plastic surgery (including 550 free flaps and 250 (super)microsurgical LVA-procedure). See page 28 line 214-215. Furthermore, we have added in the discussion that the surgeries in this study were performed by a single microsurgeon. Page 20, line 44. Furthermore, we have stated that for future research, assessment of reproducibility amongst other surgeons is of need. Page 22, lines 92-94.

An important point of clarification: Does the time needed to perform LVA using the robotic platform include time for robot setup, placement, and calibration? If not, this is an important point worthy of inclusion/discussion.

Reply: The time to perform the anastomosis is the actual time to complete the anastomosis (for robotic 25 ± 6 minutes, and for manual 9 ± 6 minutes) and excludes setup and calibration of the

robot. We have added a specification of the time measurements in the Methods section on page 29 lines 241-243:

Total time of the surgical procedure was defined as start of incision until end of the last skin suture. Total time of anastomosis was defined as start of the first microsurgical suture until completion of the last suture of the anastomosis.

The authors employ various tools for assessment which is commendable. The authors describe a high percentage of patients who discontinued their compression sleeve at three months. However, the authors also state that patients were “not allowed to wear compressive garments or receive manual lymphatic drainage four weeks post-operatively.” What is the standard of care for patients following this 4-week time point regarding compression therapy? Can the authors provide more information regarding their rationale for cessation of compression and detail their standard post-operative compression protocol for LVA? Without this information, statistics reported regarding discontinuation of garment use in both groups can be misleading. Moreover, were surveillance measurements taken by the same individuals at each time point?

Reply: As stated in the methods section on page 28 No manual lymph drainage therapy or compressive garments were allowed in the first four weeks postoperatively. We have edited on the same page (lines 232-233) that this concerns standard postoperative care in our center for LVA procedures. The rationale is that we want to protect the superficial supermicrosurgical anastomosis in the subcutaneous fat of the forearm.

After this four-week period, patients can decide themselves to use compression garments or restart manual lymph drainage. This protocol is based on expert opinion, there currently is no level one evidence in literature on this subject yet.

Surveillance measurements were taken by the same individual at each time point. We have added this in text. Page 24 line 140.

It would be helpful to have more information regarding the overall standard protocol for patient surveillance at the institution for patients undergoing LVA.

Reply: The standard protocol for patient surveillance in our center is the same as the protocol reported in the reported study.

The inclusion of two tools to validate anastomoses is commendable. As a feasibility study, information regarding the time needed for raters to complete their assessments would be informative for other institutions interested in this robotic platform and evaluative techniques used (including barriers faced).

The authors are to be commended on the first-in-human study of robotic-assisted super microsurgery. This manuscript details a successful attempt by a single surgeon to use a novel platform in performing LVA's with generally similar outcome measures between groups. Further the authors include the multiple outcome metrics used to assess: anastomoses number and quality, operative time, and patient outcome measures used. Additional information is needed for the reader to assess whether or not this novel platform can be implemented on a broader scale with similar success.

Reply: Thank you for this suggestion. Unfortunately for this study we did not record the time that the observers needed for rating the videos of the anastomosis. For our follow-up we will certainly take this into account.

Reviewer 3

Original text:

I would like to start by congratulating the authors on their paper. It is well written, addresses an important need, and as with all new technologies, authors can be excused for being overly enthusiastic about their new technology.

I will start with more general comments and end with some specifics.

Given that this is a first study in human, and it is a comparison between manual versus robotic surgery, the difference between the two approaches needs a more elaborate discussion. The differences are not just limited to surgical time, or overall scores. In supplementary table 1 (which should be part of the main article), we note that the patency of robotic anastomosis is less than that of manual surgery, that bleeding control is less and of equal importance, the number of "achieved" anastomosis per patient was higher for manual rather than robotic procedures (robotic 1.75 vs manual 2.2). The importance of these differences between the two surgical approach should be discussed. Overall surgical time can as much be surgeon related (tiredness) or available OR time for the procedure.

Reply: Thank you for your suggestions. On page 21 of the Discussion, lines 63-68, we elaborated on this matter. Mean SAMS and UWOMSA scores were significantly higher in the manual LVA group compared to the robot-assisted LVA group. This also accounts for the individual scores of manual versus robotic as outlined in Supplementary Table 1. However, this first report of the feasibility of robotic super microsurgery consists of small groups. Based on these small groups of robotic versus manual we cannot yet make solid conclusions relating the observed differences between the two surgical approaches. This matter will certainly be discussed in future larger studies.

In addition, the included supplementary table is to provide full transparency, but as already mentioned, based on the small population we cannot yet draw firm conclusions regarding the reported differences.

I would also appreciate a more detailed report, even if subjective, on surgical situations where the robot is superior to manual surgery - with smaller, more delicate vessels? difficult angles of approach?

Reply: We appreciate your suggestion. Unfortunately, we cannot yet report this based on our study as this pilot study lacks data to support any statement.

Patients had longer duration of edema and somewhat more advanced stages in the manual group, though these differences (small numbers) did not reach statistical significance. Nonetheless, an analysis that would have only considered patients with a stage 2a might provide a more homogeneous basis for a comparison of the two approaches. Does this amplify certain of the differences observed or the outcomes.

Reply: Thank you. We will consider this for future studies. 18 out of 20 patients were stage 2A. Indeed in retrospect we could have excluded the other stages. However, for this pilot study with small groups we focused on feasibility.

Also it is not clear how much training the surgeon had on the system prior to performing his first human surgery. I think it is important to state this explicitly in terms of hours, number of simulated cases, or sequences of an established training module, not just indicate that an experienced surgeon performed the surgery.

Reply: On page 28 lines 214-215 we added the amount of prior training, years of experience and number of (super)microsurgical procedures performed.

Specific items:

Abstract page 3 line 60: a decline in the time needed to complete an anastomosis was observed... Also provide number indicating the difference between robotic and manual surgery

Reply: In the Abstract we have added the decline of time to perform the robotic anastomosis in the abstract. Page 3 line 61.

P.8: be more explicit about the technical errors that precluded robotic surgery in two patients.

Reply: The two technical issues raised before the patient entered the operating room. One issue concerned a microsurgical instrument, which was appeared to be bend during the sterile process. Second issue concerned a software initialization process. In both cases the patient experienced no harm or inconvenience. We have added an explanation of the two technical failures in the text (page 9 line 195-201).

P.9 line 183 - you state the number of anastomosis completed, were all patent or equally patent at the end of surgery? If not indicate the number of patent anastomosis or indicate total number of attempts.

Reply: All anastomoses were patent at the end of surgery (both robotic assisted and manual). Page 10 line 217.

Line 190 - what corresponds to "total time". Does it include installation and draping or only the surgical procedure?

Reply: This is the time of the surgical procedure. The robot is setup and draped at the time the patient is called in and prepared for surgery (changing, indication of surgery location and local anesthesia). This installation time was not recorded. See additional explanation on page 29 lines 241-243.

P.10 Figure 2 does not add additional information, unless the image provides a representative image of a manual and a robotically completed anastomosis.

Reply: Figure 2 provides a representative image of a completed LVA.

P.16 figure S2, please add error bars to your graph.

Reply: We included error bars in Supplementary Figure 2. See modified figure S2 on page 18, with adjusted subtitle.

P.18 324 please discuss in more detail other parameters such as patency, number of anastomosis. Time constraints on the use of the operating room seem a more important factor than the time for achieving an anastomosis, given the difference in the number of achieved anastomosis between the two groups.

Reply: We concur there are more anastomosis in the hand group, this is only due to availability of suitable veins and/or lymphatics in these individual cases. We think this is merely coincidence, we cannot draw conclusions regarding this matter in the small compared group.

P.19 347 yes, but we are not interested in assessing the performance of the robot or optimizing its performance but in comparing outcome between robot and manual surgery, though a scale that assesses robotic performance will in the future be needed.

Reply: Thanks for your comment.

There currently exists a scale (the so-called SARMS), but this is developed for robots such as the DaVinci robot, as it encompasses camera use (see also explanation on page 21 lines 73-75). For the newly developed robot a new scale would be needed for the purpose as described in your comment. We agree that in the future larger groups a dedicated scale should be used to compare between hand and robot.

P.25 Given the journal and the novelty of your robot, I would give a more elaborate description, as well as an adequate description of the forceps on the robotic system. A clearer outline of whether this is a tele-operated robot, a co-manipulator, etc is needed. Also is there a remote center of motion? This can be provided as a supplement, at least the figures.

Reply: We have added additional information about the robotic system in the Online Methods section, page 27, lines 192-211. See also page 4 lines 81-82. Also the description of figure 1 is adapted, see page 7 and 8, lines 157-172.

The system can be combined with conventional microsurgical instruments and microscopes; we use sterile adapters, which can easily be placed into the robot, thereby enabling the installation of genuine microsurgical instruments to the robot. The aforementioned process only adds 10-15 seconds.

Please, see also reply on comment 8: The robot is setup and draped at the time the patient is called in and prepares for surgery (changing, indication of surgery location and local anesthesia). This

installation time was not recorded.

487; when you mention combined with genuine microsurgical instruments, you mean these can be fitted onto the robot? How much time does the installation and draping take. Is there no restriction to the surgical field or collaboration between assistant and head surgeons? Was this assessed during the trial?

Reply: We have added additional information about the robotic system in the Online Methods section, page 27, lines 192-211. See also page 4 lines 81-82. Also the description of figure 1 is adapted, see page 7 and 8, lines 157-172.

The system can be combined with conventional microsurgical instruments and microscopes; we use sterile adapters, which can easily be placed into the robot, thereby enabling the installation of genuine microsurgical instruments to the robot. The aforementioned process only adds 10-15 seconds.

Please, see also reply on comment 8: The robot is setup and draped at the time the patient is called in and prepares for surgery (changing, indication of surgery location and local anesthesia). This installation time was not recorded.

491: as previously stated, how much training on the robot did the surgeon have?

Reply: the microsurgeon had 20 hours of training on the MSR, she completed at least 3 anastomoses with a SAMS score of at least 3. See page 28, lines 216-218.

Marc D. de Smet, MDCM, PhD, FRCSC, FMH

REVIEWERS' COMMENTS:

Reviewer #1 (Remarks to the Author):

I feel that the relevant concerns of the first round of reviewers have been materially answered. It is suitable for publication.

Reviewer #2 (Remarks to the Author):

We thank the authors for their well thought out responses. No further revisions are recommended.

Reviewer #3 (Remarks to the Author):

My queries have been adequately answered